# Molecular Mechanisms of Mesenchymal Stem Cell-Based Therapy in Acute Kidney Injury

**DOI:** 10.3390/ijms222111406

**Published:** 2021-10-22

**Authors:** Pei-Wen Lee, Bo-Sheng Wu, Chih-Yu Yang, Oscar Kuang-Sheng Lee

**Affiliations:** 1Institute of Clinical Medicine, School of Medicine, National Yang Ming Chiao Tung University, Taipei 11221, Taiwan; pwlee1980@gmail.com (P.-W.L.); boshengw16507@gm.ym.edu.tw (B.-S.W.); 2Hong Deh Clinic, Taipei 11251, Taiwan; 3Faculty of Medicine, School of Medicine, National Yang Ming Chiao Tung University, Taipei 11221, Taiwan; 4Department of Medicine, Division of Nephrology, Taipei Veterans General Hospital, Taipei 11217, Taiwan; 5Center for Intelligent Drug Systems and Smart Bio-Devices (IDS2B), National Yang Ming Chiao Tung University, Hsinchu 30010, Taiwan; 6Stem Cell Research Center, National Yang Ming Chiao Tung University, Taipei 11221, Taiwan; 7Department of Orthopedics, China Medical University Hospital, Taichung 40447, Taiwan

**Keywords:** acute kidney injury, mesenchymal stem cells, extracellular vesicles

## Abstract

Acute kidney injury (AKI) causes a lot of harm to human health but is treated by only supportive therapy in most cases. Recent evidence shows that mesenchymal stem cells (MSCs) benefit kidney regeneration through releasing paracrine factors and extracellular vesicles (EVs) to the recipient kidney cells and are considered to be promising cellular therapy for AKI. To develop more efficient, precise therapies for AKI, we review the therapeutic mechanism of MSCs and MSC-derived EVs in AKI and look for a better understanding of molecular signaling and cellular communication between donor MSCs and recipient kidney cells. We also review recent clinical trials of MSC-EVs in AKI. This review summarizes the molecular mechanisms of MSCs’ therapeutic effects on kidney regeneration, expecting to comprehensively facilitate future clinical application for treating AKI.

## 1. Introduction

The kidney is a victim of critical systemic illnesses, and thus AKI usually develops as a complication during hospitalization and is related to increased mortality. The definition of AKI is a rapid decline of the glomerular filtration rate, leading to dysregulation of electrolytes, fluid, and waste product excretion. Additional complications include respiratory, cardiac, neurologic, and even multi-organ damage [1,2]. The prevalence and mortality of AKI in critical patients had been estimated in a multicenter, multinational study. The results showed that 5–6% of critical patients need renal replacement therapy, and the overall mortality was 58–62.6% [2]. The severity of AKI is associated with significantly accelerating mortality, hospitalization length, and excess hospital costs [3]. AKI can aggravate chronic kidney disease (CKD) progression and even lead to end-stage kidney disease [4]. Greater AKI severity, longer AKI duration, and more AKI episodes were risk factors for CKD progression and mortality in the hospital course [5]. Therefore, it is crucial to investigate more aggressive therapies for AKI to attenuate kidney tissue injury and CKD progression effectively.

The AKI classification can be categorized by different diagnostic etiologies: pre-renal cause, post-renal cause, and most importantly, intrinsic renal cause, representing the actual kidney disease [6]. The common causes of intrinsic AKI include ischemia, hypoxia, immunologic reaction, and nephrotoxins exposure causing damages to different parts of the kidney, such as glomerular and tubulointerstitial diseases [4].

After kidney injury, the tubular cells underwent transient de-differentiation with epithelial-to-mesenchymal phenotype transition, cell migration, and metastasis. Subsequent differentiation into mature tubular epithelial cells (TECs) occurred to replace apoptotic cells and repopulate the denudated tubular basement membrane. The kidney repair process was mainly regulated by microenvironment paracrine communication, including mediators such as insulin-like growth factor-1 (IGF-1), epidermal growth factor, transforming growth factor-β, vascular endothelial growth factor (VEGF), epidermal growth factor, hepatocyte growth factor (HGF), and vascular endothelial growth factor (VEGF) [7,8]. Furthermore, growth factors and G protein-coupled receptors (GPCRs) also participate in the local paracrine repairing mechanisms, and several GPCRs-medicated signaling are involved in the repair process of AKI [9]. In 2005, Cantley et al. revealed that human stem cells, e.g., renal tubule progenitor cells in the kidney interstitium, or stem cells mobilizing from bone marrow, possess a protective effect in the renal repair process and potential therapy for AKI [10].

The acronym “MSC” usually refers to two different words in different literature: mesenchymal stem cell or mesenchymal stromal cell [11,12]. The two words have different nomenclature and definition. The former refers to stem cells in the skeletal tissue which are capable of both self-renewal and differentiation. The latter refers to the various cells in the stromal tissues, including multipotent mesenchymal stromal cells (with the same acronym “MSC”), progenitors, and differentiated mature cells. The International Society for Cellular Therapy (ISCT) recommended minimal criteria to characterize multipotent mesenchymal stromal cells: (1) plastic-adherent in standard culture conditions; (2) positive for CD105, CD73, and CD90, and negative of CD45, CD34, CD14 or CD11b, CD79a or CD19 and HLA-DR surface molecules; (3) the differentiation ability into mesodermal lineages: osteoblasts, adipocytes, myocytes, and chondrocytes in vitro [13]. The multipotent mesenchymal stromal cells have the capability of self-renewal and multidirectional differentiations as mesenchymal stem cells do [14,15,16]. Therefore, the literature included in this article was from studies of either mesenchymal stem cells or multipotent mesenchymal stromal cells.

MSCs are present in multiple tissues and may be mobilized from the bone arrows where they are stored within perivascular niches. The MSCs have migratory homing capacities for cell regeneration during tissue injuries, inflammatory reactions, or certain cancers [17,18,19]. In addition, several MSC functions, including differentiation ability, proangiogenic, and immune-modulatory properties, make them ideal candidates for renal regenerative therapy [20,21,22]. How MSCs affect the recipient cells and result in their regenerative and immunomodulatory functions and the responsible signaling pathways were not fully understood [23,24].

MSCs are barely detectable in the recovery tissue, and this phenomenon suggests paracrine effects rather than MSCs per se are responsible for the tissue regeneration [23]. MSCs have the paracrine abilities to release different small particles, such as EVs, for direct intercellular communication [25]. EVs are derived from donor cells, lipid bilayer-delimited microparticles (MPs), and work in the microenvironment to carry a cargo of proteins, lipids, messenger RNAs (mRNAs), microRNAs (miRNAs), long non-coding RNAs (lncRNAs), and other organelles from donor stem cells to recipient cells [26]. EVs could be classified by size, biogenesis, releasing process, and function into three subgroups: exosomes (Exos), microvesicles (MVs), and apoptotic bodies [25,27,28]. Exos size from 30 to 150 nm and are secreted by exocytosis. MVs are larger, ranging from 100 to 1000 nm, and are budding from the donor cell. In multicellular organisms, MVs and other EVs are found both in tissues. Apoptotic bodies are larger than Exos and MVs and are apoptosis-related fragments derived from apoptotic cells [27]. The dominant hypothesis is that MSC-derived EVs may transfer active biologically informative particles, e.g., proteins, lipids, nucleic acids, as transcriptive encoding mediators in the intercellular communication and gene modulation for therapeutic purposes [23].

## 2. Mesenchymal Stem Cell-Based Therapy in AKI

### 2.1. Different Stem Cells Sources in AKI

Mechanisms involved in self-renewal, differentiation, and repair after kidney injury is still unclear. Evidence showed that stem cell therapies have protective effects on AKI [29]. It was found that the resident renal tubular cells in the injured denudated tissue expressed CD133, an important stem cell marker and a molecule for Wnt/beta-catenin signaling for cell proliferation and tissue repair. Embryonic renal cells express CD133 and maintain uneven expression on the scattered tubular cells in different segments. These resident scattered renal progenitor CD133^+^ cells act like stem cells, capable of self-renewal and differentiation [30]. Besides, endothelial progenitor cells (EPCs) from bone marrow can also act as a source of regenerative progenitor cells for AKI via the surface protein expression of L-selectin, an adhesion molecule that promotes EPCs migration to injured tissue [31]. Different stem cell sources, such as EPCs [31,32], human liver stem cells (HLSCs) [33], and MSCs [34,35], had demonstrated protective effects in murine AKI models. Among different stem cell sources, MSCs have the most extensive preclinical evidence and can be harvested from variable tissues, e.g., bone marrow, umbilical cord, adipose tissue, etc. Herrera et al. found that exogenous MSCs migrated to the injured kidney through the interaction between MSC surface CD44 and its ligand, hyaluronic acid), which promoted kidney regeneration [17]. Ullah et al. summarized molecular mechanisms of MSC homing steps: (1) tethering by selectins, (2) activating by cytokines, (3) arresting by integrins, (4) transmigrating by matrix remodelers, (5) extravascular migrating by chemokine gradients [36]. Besides, human induced pluripotent stem cells (hiPSCs) and embryonic stem cells were both demonstrated to promote kidney regeneration [37]. Schubert et al. had demonstrated the MSC migration and homing by tracking the adipose-derived MSC (ADSC) with bioluminescence imaging (BLI) versus quantitative reverse transcription-polymerase chain reaction (qRT-PCR) in the cisplatin-induced AKI model [38]. They detected the labeled Luc-specific mRNA in the injured kidney tissue using qRT-PCR; however, they only detected Luc^+^-ADSCs in the lung but not in the kidney, suggesting that Luc-based qRT-PCR might be a better tool than BLI to track the transplanted MSCs. In 2021, Chen et al. reported a novel therapy combined with plerixafor (AMD3100, as an immunostimulant) and granulocyte-colony stimulating factor (G-CSF). The combinative therapy of AMD3100 and G-CSF could mobilize bone marrow-derived mesenchymal stem cells (BM-MSCs) to the injured tissue and ameliorate kidney function deterioration in the cisplatin-induced AKI model [39].

### 2.2. MSCs and MSC-Derived EVs Protect from Acute Tubular Injury in Different Models

EV trafficking is multidirectional cell-to-cell communication in damaged tissues and has the effect of facilitating and reprogramming regenerative cells [40]. Additionally, the EVs contained various therapeutic mediators, including cytokines, proteins, and miRNAs [41]. The application of EVs for clinical use relies on efficient EV isolation and characterization [27,42]. 

Numerous studies in various experimental AKI models demonstrated that MSCs and MSC-derived EVs had the therapeutic activities to reverse AKI via cell-to-cell paracrine communication rather than MSCs per se transdifferentiation. Gatti et al. reported that human MSC-derived EVs could prevent AKI and CKD progression by inhibiting TEC-apoptosis while promoting TEC proliferation in the ischemia-reperfusion (I/R) AKI model [43]. A similar therapeutic effect for AKI prevention had also been observed by using EVs from renal glomerular and tubular progenitor CD133^+^ cells [44]. Besides, Chen et al. found MVs extracted from Wharton’s jelly MSCs of the umbilical cord could mitigate renal fibrosis, induce cell proliferation, and inhibit cell apoptosis via ERK1/2 signal pathway to initial G2/M cell cycle arrest in the I/R AKI model [45]. MSC-derived MVs administered in the cisplatin-AKI model have been found to upregulate anti-apoptotic genes, including Bcl2, Bcl-xL, and BIRC8. The down-regulation of apoptosis genes, e.g., Casp1, Casp8, and LTA, was also noted [46]. HLSC, with MSC surface markers but without hematopoietic/endothelial markers, and HLSC-derived EVs enhanced renal tubular regeneration in a murine glycerol-induced AKI model [33]. Furthermore, human MSC-conditioned media (MSC-CM) also possess regenerative properties for tissue injury due to MSC-secreted products, such as proteins, lipids, cytokines, or EVs, etc. Overath et al. found that MSC-CM from ADSCs preincubated in a hypoxic environment contains more protective factors and had better therapeutic effects for cisplatin-induced AKI mice than ordinary MSC-CM [47]. These studies show that MSC-derived EVs can be an innovative therapy for AKI.

### 2.3. MSCs and MSC-Derived EVs Protect from Acute Glomerular Injury in Different Models

The pathophysiology of AKI is multifactorial and complex. Definite diagnosis of AKI etiologies is sometimes challenging because multiple kidney injuries may coexist. The two major causes of acute tubular damage are ischemic and nephrotoxic [4]. Another important AKI etiology is glomerular damage, including primary rapidly progressive glomerulonephritis (RPGN), or glomerulonephritis secondary to systemic lupus erythematosus, or bacterial endocarditis [48,49]. The epidemiology, clinical phenotypes, and pathophysiology are different between acute tubular and glomerular injuries [6]. We summarized MSC-derived EVs in experimental AKI models of glomerular and tubular damage and listed the potential involved factors and main effects of EV-related kidney repair (Table 1).

An experimental glomerulonephritis rat model was created by intravenous infusion of anti-Thy1.1 antibody through complement-mediated mesangial cell damage, impairing glomerular angiogenesis. Tsuda et al. found that allogeneic fetal membrane-derived MSCs (FM-MSCs) decreased urinary protein excretion by inhibiting glomerular monocyte infiltration and mesangial matrix hyperplasia histologically in the rats with anti-Thy1 glomerulonephritis. The mechanism was evident by in vitro experiment showing that FM-MSC conditioned medium contributed to the healing process in injured kidney tissue by inhibiting prostaglandin E2-dependent expression of TNF-α and monocyte chemoattractant protein 1 (MCP-1; a key chemokine for monocyte activation) [50]. In another glomerulonephritis rat model induced by Adriamycin, Zoja et al. found that multiple administrations of BM-MSCs increased VEGF expression and reduced monocyte infiltration, podocyte apoptosis, and microvascular rarefaction and subsequently attenuated glomerular sclerosis [51]. Moreover, Iseri et al. reported that human MSC-CM treatment reduced TNF-α-related proinflammatory cytokine and modulated the glomerular macrophage polarization. The shift to anti-inflammatory M2 subsequently reduced proteinuria and crescent formation in the rat AKI model with anti-glomerular basement membrane RPGN [52]. These studies demonstrated the MSCs significantly decrease proteinuria and regain renal function through various immunomodulatory pathways.

## 3. Delivered Organelles Shuttled from MSC-Derived EVs

### 3.1. Surface Proteins

EVs can act as cargoes for mediator delivery between nearby cells (autocrine or paracrine) or distant cells (endocrine) [53]. These shuttled MSC-derived EVs with surface proteins induce internalization through a specific membrane receptor-mediated mechanism on the recipient cells [54]. The EV internalization mechanisms involve protein interactions between MSCs and recipient cells that facilitate subsequent endocytosis [55]. The endocytosis of MSC-derived EVs into the recipient cells was through various mechanisms, including clathrin-dependent and clathrin-independent endocytosis. The latter had different pathways, such as caveolin-mediated endocytosis, phagocytosis, macropinocytosis, and lipid raft-mediated endocytosis [56,57].

In experimental kidney injury models, EV internalization was orchestrated by the up-regulation of adhesion molecules of injured cells [58]. This result was consistent with higher adherent intercellular cell adhesion molecule-1 (ICAM1), which promoted EV-cell fusion [58,59]. The expressed surface proteins on the MV membrane originate from the donor stem cells due to the same surface molecules characteristics [60]. Moreover, the protein and lipid components on the Exos membrane had been extensively studied, including adhesion molecules (e.g., ICAM-1, CD146), tetraspanins (e.g., CD63, CD81), and lipid rafts microdomains (e.g., lyso-bisphosphatidic acid-binding protein Alix) [61]. Bruno et al. reported that MSC-derived MVs shared the same surface adhesion molecules of MSCs they originated, CD44 and CD29 (also named as β1-integrin), which were involved in the internalization of MVs into TECs. Removal of surface proteins resulted in the inhibition of MVs and TECs incorporation [62]. Besides, Chol et al. also demonstrated kidney-derived MSC (KMSC)-derived MPs expressed the same membranous adhesion molecules as the KMSC they originated, such as CD29, CD44, CD73, α4-, α5-, and α6-integrins [63].

Figure 1 depicts the therapeutic mechanisms of MSC-EVs on kidney cell regeneration in AKI.

### 3.2. Nucleic Acid (mRNA and miRNA) Trafficking

Engraftment of EVs is a paracrine or endocrine communication between donor stem cells and recipient injured cells. EVs are cargoes containing genetic information (mRNA, miRNA), proteins, lipids, cytokines, complements, immune complexes, and other organelles [53,64]. After being internalized, EVs triggered a series of reactions to regulate M1 and M2 macrophage polarization and then modulated damaged tissue inflammation and regeneration. This effect is mainly ascribed to horizontal nucleic acid transfer, including mRNA and miRNA [58,65]. The following section outlines the nucleic acid components delivered from EVs and their potential pathophysiology in different experimental AKI models. Furthermore, we list the experimental EV factors delivered under MSC-derived EVs therapies for AKI and try to answer which factors may be promising for clinical regeneration applications in the future (Table 2).

#### 3.2.1. Delivery of miRNA

The miRNAs could regulate post-transcriptional gene expression, especially the genes involved in tissue repair [66]. Gatti et al. first demonstrated that RNase could induce the MSC-derived MV cargo degradation and MVs provide this paracrine action of intercellular horizontal transfer of mRNA and miRNA [43,79]. Moreover, Collino et al. also found that MSC-derived EVs carried miRNAs to reprogram a pro-regenerative phenotypic change. They used a specific miRNA depletion Drosha-knockdown MSCs (MSC-Dsh) and collected the EVs derived from MSC-Dsh (EV-Dsh). They sorted the miRNA profile of the EV-Dsh, and the result showed 49 miRNA down-regulation compared with the control EVs. Both The EV-Dsh and MSC-Dsh inhibited TEC regeneration in a glycerol-induced AKI model, whereas the wild-type MSC and EVs were effective for AKI recovery. Gene ontology (GO) analysis of these down-regulated genes in EV-Dsh had a correlation with extracellular matrix (ECM)-receptor, surface adhesion molecules, Wnt pathway, p53 signaling, and ECM remodeling [67].

In an in vitro I/R AKI model, Lindoso et al. incubated the PTECs with transcription inhibitor actinomycin D and then shifted these PTECs in an ATP depletion injury model. Compared to the transcription blockage cells, injured PTECs treated with MSC-EVs showed increased expression of miR-148b-3p, miR-410, miR-495, and miR-548c-5p despite transcription inhibitor actinomycin D treatment. The findings suggest that these miRNAs were directly transferred from EVs. They also found other down-regulated miRNA in PTEC, which were directly transferred from EVs or induced by EVs, including let7-a, miR-148b-3p, 375, 410, 451, 485-3p, 495, 522, 548c-3p, 548c-5p, 561, and 886-3p. GO analysis revealed these down-regulation miRNAs were related to apoptosis, cytoskeleton reorganization, and hypoxia (e.g., SHC1, SMAD4, CASP3, and 7). It suggests that these modulated miRNAs may be responsible for reducing kidney cell death through inhibiting ATP depletion [66].

Another evidence that miRNA transferred by EVs regulates kidney cells regeneration was from the study by Gu et al., who had identified that human Wharton Jelly mesenchymal stromal cells (hWJMSCs)-EVs alleviated cell apoptosis via regulating dynamin-related protein1 (DRP1) expression and mitochondria fission through a miR-30-related anti-apoptotic mechanism. The Drp1 is a GTPase to control mitochondrial dynamics, including fission and fusion, which are key steps in cell aging and apoptosis [80,81]. Their data showed that hWJMSCs-EVs had delivered and restored miR-30b/c/d in TECs, leading to anti-apoptotic effects [68].

Moreover, Zhu et al. found another possible pathway that involves EVs protecting tubular cells from apoptosis in AKI. Exos from human BM-MSCs (hBM-MSC-Exos) were demonstrated to be internalized into the renal PECs by labeling the protein markers CD9, CD63, and TSG101. The miR-199a-3p existed numerously in Exos and also overexpressed in renal PECs subsequently. hBM-MSC-Exos alleviate kidney cell apoptosis via down-regulating semaphorin 3A expression and upregulating the AKT and ERK pathways through the delivering miR-199a-3p [69]. Besides, Cao et al. found that MSC-derived Exos delivered miR-125b-5p and promoted tubular repair by repressing the p53 expression, which rescued G2/M arrest via upregulating of CDK1 and Cyclin B1, thus inhibiting cell apoptosis [71]. Zhang et al. demonstrated that human umbilical cord MSC (hucMSC)-Exos delivered miR-146b into tubular cells to treat sepsis-associated AKI, which was mediated by the inhibition of inflammatory factor NF-ƙB through decreasing interleukin-1 receptor-associated kinase expression [70]. A study by Wu et al. indicated that in a cisplatin-AKI mouse model, BM-MSCs upregulated the miR-146a-5p/Tfdp2 axis in TECs, and Tfdp2, as a cofactor to control cell cycle and regulate c-Myc gene expression, was important for renal function stabilization [82].

#### 3.2.2. Delivery of mRNA

Evidence shows that miRNA and mRNA delivered by EVs cargo is responsible for the effectiveness of MSC-mediated therapy for AKI. They identified that mRNAs from MSC-EVs are delivered into recipient cells for intercellular communication. Bruno et al. found that RNase eliminated the MV effect on kidney regeneration in the glycerol-induced AKI model. They hypothesized RNAs exert their impacts on the regulation of gene transcription, cell proliferation, and immunomodulation. More understanding of MV-RNA extracts was explored by newly developed quantitative real-time PCR. Their findings indicated that MVs delivered specific groups of cellular mRNAs as reporter miRNAs, including upregulated anti-apoptotic and down-regulated apoptosis genes [46,62]. 

Ragni et al. demonstrated that *IL-10* mRNA could be delivered from different sources, such as BM-MSC-EVs and ucMSC-EVs, and then be internalized into recipient renal PTEC (HKC-8) in the cisplatin-AKI model in vivo and in vitro. The renal TECs didn’t express IL-10 initially and obtained the *IL-10* mRNA and de novo IL-10 protein after co-culture with BM-MSC-EVs and ucMSC-EVs. These findings suggest EVs enhanced IL-10/IL-10R1R2 anti-inflammatory pathway via translated *IL-10* mRNA, indicating that the therapeutic mRNAs in EVs provided a promising therapy for AKI [23]. In another similar paradigm, Tomasoni et al. found that hBM-MSC-derived MPs and Exos transferred IGF-1 receptor (IGF-1R) mRNA to cisplatin-damaged PTECs, and IGF-1R protein was expressed subsequently, which was involved in the process of inflammatory and PTEC proliferation [72]. HGF is a paracrine and pleiotropic protein and acts as a cytokine to bind to the proto-oncogenic c-Met receptor on many kinds of cells, primarily of epithelial origin. The HGF receptor, c-Met, on the renal tubular epithelium could activate a tyrosine kinase signaling cascade and then promote tissue repair and regeneration in AKI via altering their gene expression of regulating cell growth and morphogenesis. MSC-derived EVs delivered *HGF* mRNA into the TECs, where HGF protein and transforming growth factor-beta1 (TGF-β1) protein synthesis were induced. These gene regulation and protein expression could reduce renal fibrosis and scar formation by disturbing tubular epithelial-mesenchymal transition (EMT), activating the Erk1/2 signal, and facilitating TEC de-differentiation and regeneration [73,74,83]. Choi et al. confirmed that KMSC-derived MPs enriched in *VEGF-A*, *bFGF*, and *IGF-1* mRNA is a major player in cell proliferation, anti-apoptotic, and angiogenic effects in the I/R AKI model in vitro and in vivo [63]. Zhang et al. created three groups of hucMSCs, including normal hucMSCs, Oct-4-overexpression HUMSCs and Oct-4-knockdown HUMSCs, to confirm the hucMSCs-EV could transport *Oct-4* mRNA to renal TECs, where *Oct-4* mRNA abrogated EMT and fibrogenesis by inhibiting Snail gene activation, which is an important transcription factor to induce EMT phenomenon in fibroblasts [75,84].

Solid evidence demonstrated that mRNA transferred from MSC-MVs protects both renal tubular cells and peritubular capillary endothelial cells. In most in vivo models, MVs were internalized at the apical membrane of TECs. Bruno et al. confirmed that labeled MVs were detectable within the capillary endothelial cells 1 h after MV administration and then detected within TECs 2 h later. This suggests that MVs could reach the basolateral membrane of TEC through the peritubular capillary endothelial cells [62].

#### 3.2.3. Delivery of lncRNA

The lncRNA is a kind of non-coding RNA with more than 200 nucleotides in length. It has two primary functions, interacting with other proteins to assemble ribonucleoprotein complexes or competing with other non-coding RNAs to regulate gene expression. RNA transcription, epigenetic regulation, DNA stability, and aging process modulation are regulated by lncRNA [85,86,87]. Unlike other single-strand RNA, lncRNAs act as modular scaffolds with more stable structures and highly specific spatial and temporal construction [88,89]. There are five categories of lncRNAs according to genomic locations, RNA transcriptive direction, and interaction with protein-coding genes: enhancer RNAs, small nucleolar RNA hosts, intergenic transcripts (located intergenically from both strands), sense-lncRNA (overlapping with exons from protein-coding genes in sense orientation), and antisense-lncRNA (located in the antisense strand orientation of protein-coding genes) [88,90]. Moreover, lncRNAs play important roles by regulating gene expression in various biological activities and disease progression, such as autoimmune disorders, coronary artery diseases, cancers, and neurological diseases [85,91,92,93].

Although past studies on lncRNA activity focused mainly on cancer, they may also be involved in stem cell therapy because of the chromatin regulation ability. The EV cargo transfers various paracrine mediators. One of them is lncRNAs that are increasingly being studied about their biological functions and may help treat diseases and injuries across all organ systems [26]. Copper et al. reported that human ADSC-Exos and CM without stem cells both contained abundant lncRNA *MALAT1* (metastasis-associated lung adenocarcinoma transcript 1). When the *MALAT1* enriched Exos and CM were used to treat ischemic excisional wounds in a rat model, the wound healing was improved via dermal fibroblast migration and increased angiogenesis [94]. Besides, Zhu et al. found that hucMSC-Exos could deliver lncRNA *MALAT1* into rat cardiomyocytes and improve cardiac repair by activating the NF-κB/TNF-α signaling pathway in the inflammatory heart tissue. They evaluated the cardiac function of the D-galactose-treated mice 42 days after Exos injection, and the result showed Exos induce cardiac repair and prevent aging-induced cardiac dysfunction [95]. Liu et al. showed that MSC-Exos contained lncRNA *KLF3-AS1* improved chondrocyte proliferation and cartilage repair in a collagenase-induced osteoarthritis model [96]. Hou et al. reported that plasmid with over-amplification of lncRNA *H19* could promote MSC proliferation and survival in the hypoxia/serum deprivation model through the mechanism of competition with miR-199a-5p with subsequent up-regulation of VEGF-A expression [97]. Finally, Jin et al. demonstrated that hADSC-EVs released lncRNA *H19* and had a therapeutic effect for hepatocyte recovery in rats with acute liver failure [98].

Focusing on the lncRNA in the AKI field, there is increasing evidence of lncRNA involvement in AKI. Lorenzen et al. first identified a novel intronic antisense lncRNA, named TrAnscript Predicting Survival in AKI (TapSAKI), which could be found in kidney biopsy samples, especially enriched in hypoxic TECs, and also circulated in the blood of AKI patients [99]. The baseline concentrations of circulating TapSAKI lncRNA increased with the disease severity and correlated with the 28-day survival rate. The TapSAKI lncRNA was up-regulated in lipopolysaccharide (LPS)-induced HK-2 cell injury model, and knockdown of TapSAKI lncRNA could diminish kidney cell injury through several signaling pathways, including miR-205/Interferon regulatory factor 3 (IRF3) pathways, miR-22/PTEN/TLR4/NF-κB pathway [100,101]. The increasing evidence about lncRNAs in injured tissue indicates a new player of regeneration therapy for AKI. 

### 3.3. Protein Trafficking

EV effects on recipient cells are not only mediated by nucleic acid but also by functional protein trafficking. Wang et al. demonstrated that 14-3-3ζ proteins transported by hucMSC-derived Exos might prevent HK-2 cells from cisplatin injury through involvement in the protective cell digestion process to overcome environmental stress [77]. The binding protein 14-3-3 family could regulate the autophagy process through different signal pathways, such as MAPK, PI3K, and mTOR [102]. Tseng et al. also found that MSCs stimulate the autophagy-related protein expression, such as LC3B, Atg5, and Beclin, in the I/R AKI rat model and then improved AKI recovery [78]. Yuan et al. also identified that hiPSC-derived MSCs secreted the paracrine signal protein, specificity protein (SP1), into recipient TECS, where the SP1 upregulated the sphingosine kinase 1 (SK1) to induce sphinganine-1-phosphate (S1P) formation, which was necessary for the anti-necroptosis effect. However, SP1 mRNA levels remain unchanged, suggesting SP1 proteins from EVs were directly internalized by target cells [76].

## 4. Clinical Trials

Positive preclinical investigations encouraged the phases I and II clinical trials to explore MSC-based novel therapy for AKI [103]. In the database of ClinicalTrials.gov, six clinical trials are currently undergoing or completed. These clinical trials target the AKI field and focus on the safety and efficacy of stem cell therapy following cardiac surgery, solid organ cancer patients, COVID-19 subjects, and those who are receiving continuous renal replacement therapy [104]. Table 3 summarizes clinical trials testing the efficacy of stem cell therapies in AKI.

A phase I trial was completed in 2013 (NCT00733876) to evaluate the safety and efficacy of BM-MSC treatment in 16 high-AKI-risk subjects who received on-pump cardiac surgery. The results showed that MSC administration is a safe and protective therapy for AKI [105,106,107]. However, these effects were not validated in the subsequent phase II study (NTC01602328) that enrolled 156 patients. The patients displayed untimely signs of AKI after cardiac surgery. These patients were randomized to intra-aortic injection of allogeneic MSCs or standard treatment. The patients receiving allogeneic MSC were harmless and tolerated but could not improve AKI or mortality. Hence, this clinical trial was terminated in 2014 [108]. In another phase I clinical trial (NCT01275612), a single-dose intravenous injection of allogeneic BM-MSCs was administered in subjects who suffered from AKI due to cisplatin treatment for solid organ cancer, and they were followed up for one month after allogeneic BM-MSCs infusion. Primary and secondary endpoints included the rate of renal function decline and urinary injury markers. However, patients evaluated so far could not be enrolled because they did not meet the primary criterion of AKI provided by the study protocol. Therefore, this study was withdrawn in 2018.

There are three ongoing studies. A phase I/II study (NCT03015623) was started in 2017 and planned to enroll 24 participants with AKI receiving continuous renal replacement therapy and treat them with SBI-101 treatment for up to 24 h. SBI-101 is a biologic combinational device with two constituents: allogeneic human MSCs and an FDA-approved plasmapheresis machine, designed to alleviate inflammation and facilitate tissue repair [109]. The study is active but not yet recruiting. Regarding the NCT03015623 clinical trial, another phase I/II study (NCT04445220) enrolled 22 COVID-19 subjects with AKI who received renal replacement therapy in 2020. The patients were treated with SBI-101 therapy for up to 24 h [110]. Finally, an ongoing phase I/II clinical trial (NCT04194671) explores novel therapies for intravenous administration of MSC on days 0 and 7. This trial evaluates the safety and efficacy of BM-MSC therapy and renal function change in patients within 28 days after MSC therapy.

In summary, abundant optimistic preclinical studies encouraged aggressive pilot clinical trials to explore the new MSCs-based therapies. However, stem cell therapies still face challenges, including the risk of rejection, fibrogenesis, tumorigenesis, and embolization [40,111].

## 5. Conclusions and Future Perspectives

AKI remains one of the leading public health challenges and is a mortality predictor for critical patients. Currently, there are no efficient therapies AKI [112]. PTECs death is the most general reason for AKI and frequently occurs due to ischemia or nephrotoxins [113]. Several clinical trials are ongoing or completed using MSCs for AKI, and the results showed that MSCs as a regenerative therapy is promising. In summary, MSCs secrete specific nanoparticles such as EVs and MVs loaded with nucleic acids or nucleic acid-binding proteins. The nanoparticles are internalized into recipient cells and may act as a purposeful modulator to the target nucleic acids or target protein [23].

Compared to MSC per se, MSC-derived EVs are more diminutive in size, acellular, higher stability, more biocompatibility, and lower toxicity. They are considered to be appropriate to be MSCs-based therapy. There are many ordeals for clinical practice of MSC-derived EV therapy, including source, storage, delivery system, administration route, safety, and long-term effects [111]. Different methods for freezing, storage, and isolating EVs will influence their purity, concentration, activity, and therapeutic effects in standard protocols for EV studies. We believe more studies should be designed to identify reliable biorepositories of EV operation management [111,114,115]. Another possible solution to EV instability and heterogeneity is to create artificial vesicles with more homogeneity [116]. Moreover, EVs affect gene and transcription modulation in the target cell, and the long-term impact and risk of these stem cell-based therapy requires more studies [111].

In the clinical practice of EVs therapy for AKI, the ideal prescription of EVs, such as different dose responses and optimal intervals between multiple doses, should be established in more clinical trials [115,117]. Unfortunately, when MSCs are injected intravenously, they disappear quickly due to lung trapping [118,119]. Scientists tried to create different MSC infusion routes to bypass the lungs and elongate the MSCs’ lifespan [120]. The currently optimal route might be intrarenal administration [117], which is invasive and inappropriate for clinical use. Therefore, more comprehensive investigations of MSC-derived EV applications should be performed to develop safe and effective therapies for AKI.

## Figures and Tables

**Figure 1 ijms-22-11406-f001:**
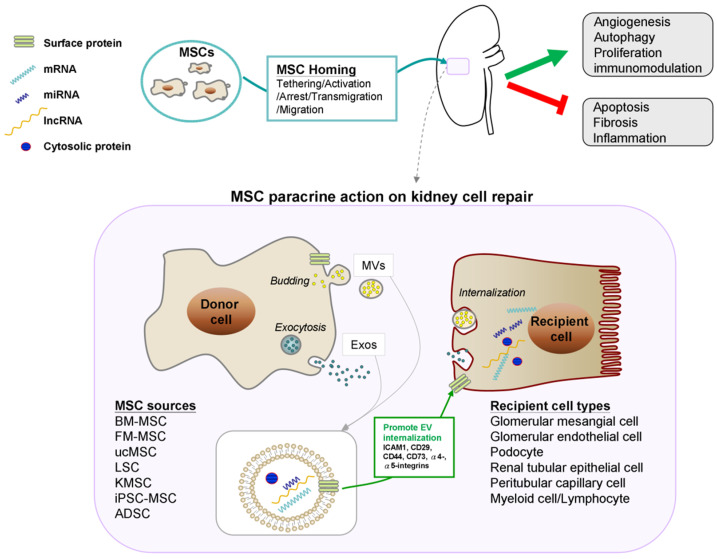
Schematic representation of the kidney regeneration by mesenchymal stem cell-derived extracellular vesicles in acute kidney injury. Abbreviations: ADSC, adipose-derived mesenchymal stem cell; BM-MSC, bone marrow-derived mesenchymal stem cell; FM-MSC, fetal membrane-derived MSCs; iPSC-MSCs, induced pluripotent stem cell-derived MSCs; ucMSC, umbilical cord-derived mesenchymal stem cell; KMSC, kidney-derived mesenchymal stem cell; LSC, liver stem cell.

**Table 1 ijms-22-11406-t001:** MSC-derived EVs in experimental models of AKI and the potential mechanisms of EV-induced renal tissue repair.

Histology	Authors/Year Reference	EV Sources	EV Types	Experimental Model	Species	EV Factors	Molecular Response	Functional Modulation
Acute Tubular Injury	Herrera et al., 2007 [17]	BM-MSCs	NM	In vitro/in vivo, glycerol-induced AKI	Mouse	NM	↑CD44 and hyaluronic acid (major ligand of CD44) interactions	↑exogenous MSC migration and homing
	Gatti et al., 2011 [43]	BM-MSCs	MVs	In vivo, I/R induced acute tubular injury	Rat	NM	NM	↓tubular cell apoptosis, ↑TEC proliferation
	Bruno et al., 2012 [46]	BM-MSCs	MVs	In vitro/in vivo, cisplatin-induced acute tubular injury	Mouse	Human POLR2E mRNA	↑anti-apoptotic genes, *Bcl-xL*, *Bcl2*, and *BIRC8*, ↓apoptosis genes, *Casp1*, *Casp8*, and *LTA*	↑renal function, morphology, and survival
	Mb et al., 2014 [33]	hLSCs	NM	In vitro/in vivo, intra-muscle glycerol induced AKI	Mouse	NM	↑PCNA expression	↑tubular cell proliferation, ↑renal function, ↑morphology
	Chen et al., 2017 [45]	hWJMSCs	MVs	In vitro/in vivo, I/R-induced renal fibrosis	Rat	NM	↑ERK1/2 signaling ↓EMT–related protein, TGF-β1 ↑cell cycle-related proteins, CDK 1 and CyclinB1	↑proliferation, ↓apoptosis, ↓collagen deposition, ↑cells in G2/M cell cycle, ↓fibrosis, ↓EMT
	Ranghino et al., 2017 [44]	Gl-MSCs T-CD133^+^ cells	Gl-MSC-EVs T-CD133^+^-EVs	In vivo, I/R induced acute tubular injury	Mouse	62 group of miRNAs	NM	↑TEC proliferation
	Overath et al., 2016 [47]	ADSC-pCM	pCM	In vitro/in vivo, cisplatin-induced acute tubular injury	Mouse	64 expressed proteins	↓inflammatory cytokines, IL-1β, IL-6	↑ survival ↓ serum Cr and N-GAL
Acute Glomerular Injury	Tsuda et al., 2010 [50]	FM-MSCs	NM	In vitro/in vivo, anti-Thy1 nephritis	rats	NM	↓TNF and MCP-1 through a PGE2-dependent mechanism.	↓Proteinuria ↓mesangial matrix/cell proliferation, ↓glomerular monocyte/macrophage infiltration,
	Zoja et al., 2012 [51]	BM-MSCs	NM	In vitro/in vivo, Adriamycin-induced crescentic nephritis	rats	NM	↑VEGF expression ↑nephrin and CD2AP	↓monocyte infiltration, ↓podocyte apoptosis, ↓microvascular rarefaction
	Iseri et al., 2016 [52]	hMSC-CM	CM	In vitro/in vivo, anti-glomerular basement membrane nephritis	rats	NM	↓proinflammatory cytokines TNF-α, IL-1-β, MCP-1, and IL-6	↑M2 macrophage polarization, ↓proteinuria and crescent formation

Abbreviations: ADSC-pCM, adipose-derived MSC-preconditioned media; AKI, acute kidney injury; BM-MSCs, bone marrow-derived MSCs; CDK, cyclin-dependent protein kinase; EMT, epithelial-mesenchymal transition; EVs, extracellular vesicles; FM-MSCs, fetal membrane-derived MSCs; Gl-MSCs, MSCs derived from human glomeruli; I/R, ischemia/reperfusion; hLSCs, human liver stem cells; hMSC-CM, human MSC-conditioned media; hWJMSCs, human Wharton’s jelly-MSCs; IGF-1R, insulin-like growth factor-1 receptor; MAC, membrane attack complex; MCP-1, monocyte chemoattractant protein 1; MSC, mesenchymal stem cell; MVs, microvesicles; N-GAL, neutrophil gelatinase-associated lipocalin; NM, not mentioned; PCNA, proliferating cell nuclear antigen; T-CD133^+^ cells, T-CD133^+^ progenitor cells from human renal tubules tissue.

**Table 2 ijms-22-11406-t002:** Experimental EV factors delivered under MSC-derived EVs therapies of AKI.

Substances Delivered	Authors/Year, Reference	EV Sources	EV Types	Experimental AKI Type	Species	EV Factors	Molecular Response	Functional Modulation
Delivery of miRNA	Lindoso et al., 2014 [66]	hMSCs	NM	H/R of PTECs in ATP depletion model	PTECs (HK-2)	20 miRNAs ( such as miR-222, miR-145, etc)	↓coding-mRNAs: *CASP-3*, *CASP-7*, *SHC1* and *SMAD4*	↓cell death by apoptosis or hypoxia
	Collino et al., 2015 [67]	BM-MSCs	NM	Glycerol	Mouse	8 miRNA families (miR-483-5p, miR-191, miR-283p, miR-744, miR-423-5p, miR-24, miR-129-3p, miR-148a)	↑genes with fatty acid metabolism, complement, and coagulation cascades ↓genes with inflammation, and adhesion molecules	↑Proregeneration, ↓hyaline casts, ↓tubular necrosis, ↓tubular damage markers: lipocalin2 and fibrinogen subunits
	Gu et al., 2016 [68]	hWJMSCs	NM	I/R by unilateral nephrectomy	Rat	miR-30	↓DRP1 expression	↓mitochondrial fission
	Zhu et al., 2019 [69]	BM-MSCs	Exos	I/R	Mouse	miR-199a-3p	↓semaphorin 3A, ↑AKT and ERK pathways	↓cell apoptosis
	Zhang et al., 2020 [70]	hucMSCs	Exos	Sepsis model through cecal ligation	Mouse	miRNA-146b	↓IRAK1 expression, ↑NF-κB activity	↑survival and kidney function
	Cao et al., 2021 [71]	hucMSCs	Exos	I/R	Mouse	miR-125b-5p	↓p53 protein, ↑CDK1 and Cyclin B1, ↓apoptosis-related proteins, Bax and cleaved-caspase-3, ↑anti-apoptosis protein, Bcl-2	↑proliferative TECs, ↓G2/M cell cycle arrest and apoptosis of TECs
Delivery of mRNA	Bruno et al., 2009 [62]	BM-MSCs	MVs	Glycerol	Mouse	*Human POLR2E* mRNA	↑cytoplasmic POLR2E protein, ↑cytoplasmic and nuclear SUMO-1 protein	↑proliferative and anti-apoptotic effects
	Bruno et al., 2012 [46]	BM-MSCs	MVs	Cisplatin	Mouse	*Human POLR2E* mRNA	↑anti-apoptotic genes, *Bcl-xL*, *Bcl2*, and *BIRC8*, ↓apoptosis genes, *Casp1*, *Casp8*, and *LTA*	↑renal function, morphology, and survival
	Tomasoni et al., 2013 [72]	BM-MSCs	Exos	Cisplatin	PTECs (HK2)	*IGF-1R* mRNA	↑*IGF-1R*-corresponding protein, IGF-1R	↑sensitivity to IGF-1, ↑PTEC proliferation
	Du et al., 2013 [73]	WJ-MSCs	NM	I/R by renal pedicle ligation	Rat	*Human* HGF mRNA	↑HGF protein expression, ↑TGF-β1, ↓α-SMA/E-cadherin	↓renal fibrosis ↓tubular EMT ↓renal fibrosis
	Choi et al., 2014 [63]	KMSCs	MVs	I/R	Mouse	*VEGF-A*, *IGF-1*, and *FGF* mRNA	↑PCNA, ↓CD 31	↑cell proliferation, ↑angiogenesis
	Ju et al., 2015 [74]	hucMSCs	MVs	I/R by renal pedicle ligation	Rat	*Human HGF* mRNA	↑HGF protein expression, ↑ERK1/2 signaling activation	↑TEC de-differentiation ↓apoptosis
	Ragni et al., 2017 [23]	BM-MSCs and hucMSCs	EVs	Cisplatin	PTECs (HKC8)	*IL-10* mRNA	↑*IL-10*-corresponding protein, IL-10, in PTECs	↑rescue AKI
	Zhang et al., 2020 [75]	hucMSCs	NM	I/R by unilateral nephrectomy	Mouse	*Oct-4* mRNA	↓Snail expression, ↓α-SMA	↓EMT, ↓apoptosis ↑proliferation
Delivery of proteins	Yuan et al., 2017 [76]	iPSC-MSCs	EVs	I/R by renal pedicle ligation	Rat	SP1	↑SP1–SK1–S1P signaling pathway, ↑SK1, ↑S1P	↓necroptosis
	Wang et al., 2018 [77]	hucMSCs	Exos	Cisplatin	PTECs (HK2)	14-3-3ζ	↑PCNA	↑autophagy
	Tseng et al., 2021 [78]	BM-MSCs	NM	I/R by unilateral nephrectomy	Rat	LC3B, Atg5, and Beclin 1	↓proinflammatory IL-1β, pro-apoptotic Bax, caspase 3 ↑autophagy-related LC3B, Atg5, and Beclin 1	↓macrophage infiltration ↓Tubular apoptosis, ↑tubular proliferation,

Abbreviations: AKI, acute kidney injury; α-SMA,α-smooth muscle actin; BM-MSCs, bone marrow-derived MSCs; CDK1, cyclin-dependent kinases 1; DRP1, dynamin-related protein 1; EMT, epithelial-mesenchymal transition; ERK, extracellular-signal-regulated kinase; EVs, extracellular vesicles; Exos, exosomes; HGF, hepatocyte growth factor; H/R, hypoxia/reoxygenation; hWJMSCs, human Wharton’s jelly-MSCs; hucMSCs, human umbilical cord MSCs; I/R, ischemia/reperfusion; IGF-1R, insulin-like growth factor-1 receptor; iPSC-MSCs, induced pluripotent stem cell-derived MSCs; IRAK1, interleukin-1 receptor-associated kinase; KMSC, kidney-derived MSCs; MSC, mesenchymal stem cell; MVs, microvesicles; NM, not mentioned; PCNA, proliferating cell nuclear antigen; PTECs, proximal tubular epithelial cells; S1P, sphinganine-1-phosphate; SK1, sphingosine kinase 1; SP1, specificity protein 1.

**Table 3 ijms-22-11406-t003:** Clinical trials testing the efficacy of stem cell therapy in AKI.

ID/Reference	Study Title	Conditions	Interventions	Status	Start and Complete Date	Link
NCT00733876 [105,106,107]	Allogeneic Multipotent Stromal Cell Treatment for Acute Kidney Injury Following Cardiac Surgery	Acute renal tubular necrosis	Biological: MSC administration	Completed	August 2008~October 2013	https://clinicaltrials.gov/ct2/show/NCT00733876 (accessed on 1 June 2021)
NCT01275612	Mesenchymal Stem Cells in Cisplatin-Induced Acute Renal Failure In Patients With Solid Organ Cancers	Solid tumors AKI	Biological: MSC infusion	Withdrawn	November 2010~19 March 2018	https://clinicaltrials.gov/ct2/show/NCT01275612 (accessed on 1 June 2021)
NCT01602328	A Study to Evaluate the Safety and Efficacy of AC607 for the Treatment of Kidney Injury in Cardiac Surgery Subjects	AKI	Biological: AC607 Biological: Vehicle Only	Terminated	June 2012~August 2014	https://clinicaltrials.gov/ct2/show/NCT01602328 (accessed on 1 June 2021)
NCT03015623	A Study of Cell Therapy for Subjects With Acute Kidney Injury Who Are Receiving Continuous Renal Replacement Therapy	AKI	Biological: SBI-101 Device: Sham	Active, not recruiting	June 2017~December 2021	https://clinicaltrials.gov/ct2/show/NCT03015623 (accessed on 1 June 2021)
NCT04445220	A Study of Cell Therapy in COVID-19 Subjects With Acute Kidney Injury Who Are Receiving Renal Replacement Therapy	COVID-19 AKI	Biological: SBI-101	Recruiting	November 2020~December 2021	https://clinicaltrials.gov/ct2/show/NCT04445220 (accessed on 1 June 2021)
NCT04194671	Clinical Trial of Mesenchymal Stem Cells in the Treatment of Severe Acute Kidney Injury	AKI	Biological: MSC Other: Saline	Not yet recruiting	31 January 2020~31 December 2022	https://clinicaltrials.gov/ct2/show/NCT04194671 (accessed on 1 June 2021)

## Data Availability

Not applicable.

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
