# Peer review of "Molecular Mechanisms of Mesenchymal Stem Cell-Based Therapy in Acute Kidney Injury"

_ijms, 2021, doi:10.3390/ijms222111406_

Round 1

Reviewer 1 Report

The literature review is comprehensive and well organized.
However, it is only accessible to a specialized audience.
The last chapter on potential clinical applications reflects the difficulties to apply these therapies in clinical situtation.

Author Response

Point-by-point responses to the reviewer (Manuscript ID: ijms-1344005)

Reviewer #1:

1. The literature review is comprehensive and well organized.

However, it is only accessible to a specialized audience.

The last chapter on potential clinical applications reflects the difficulties to apply these therapies in clinical situation.

=> Responses: We truly appreciate this valuable comment and positive response. As you mentioned, stem cell therapy for AKI is a novel research field with little understanding so far. We wish there will be more and more investigations. We will continue our efforts and look forward to its clinical use.

Reviewer 2 Report

In this review, the authors summarize the molecular mechanisms of the therapeutic effects of MSCs in the Acute kidney injury and its renal regeneration.  MSCs-based regenerative therapy can be useful in the future clinical regeneration for the treatment of AKI.

In Introduction section, the authors described Mesenchymal stem cell (MSC), also named as mesenchymal stromal cell, is one of the important multipotent, non-hematopoietic adult stem cells with the differential abilities to several cell lines, including osteoblasts, chondrocytes, muscle cells, and fat cells,  and then provide therapeutic effect with the power of self-renewal and differentiation” (lanes 61-64).

The authors must provide a clear definition of MSCs. Mesenchymal stromal cells, present in the stromal component of several tissues represent a heterogeneous population, including multipotent stem cells, progenitors, and differentiated cells (https://doi.org/10.1007/s12015-021-10231-w).  Therefore, only a fraction of the population (multipotent stem cells and progenitors) exhibit the ability of self-renewal and multidirectional differentiation into osteocyte, chondrocyte and adipocytes.

Although, definitive data supporting the stemness of this heterogeneous cell population were not provided, it has been hypothesized that a stromal stem cell population may be present in the stromal component of the bone marrow microenvironment and in that of other tissues; thus, the term “mesenchymal stem cell” should be restricted to this population of mesenchymal cells 

Please revise the correct markers that characterize the MSCs (lanes 66-67) (http://dx.doi.org/ 10.1002/stem.1681 https://doi.org/10.1007/s12015-021-10231-w, https://doi.org/10.1016/j.jcyt.2020.07.003).

Lane 185 please change surface proteins with Surface proteins

As  suggestion, a curated list of experimentally-supported associations between lncRNAs and diseases can be found at https://www.cuilab.cn/lncrnadisease

Author Response

Point-by-point responses to the reviewer (Manuscript ID: ijms-1344005)

Reviewer #2:

In this review, the authors summarize the molecular mechanisms of the therapeutic effects of MSCs in the acute kidney injury and its renal regeneration. MSCs-based regenerative therapy can be useful in the future clinical regeneration for the treatment of AKI.

1. In Introduction section, the authors described Mesenchymal stem cell (MSC), also named as mesenchymal stromal cell, is one of the important multipotent, non-hematopoietic adult stem cells with the differential abilities to several cell lines, including osteoblasts, chondrocytes, muscle cells, and fat cells, and then provide therapeutic effect with the power of self-renewal and differentiation” (lines 61-64)

The authors must provide a clear definition of MSCs. Mesenchymal stromal cells, present in the stromal component of several tissues represent a heterogeneous population, including multipotent stem cells, progenitors, and differentiated cells (https://doi.org/10.1007/s12015-021-10231-w).  Therefore, only a fraction of the population (multipotent stem cells and progenitors) exhibits the ability of self-renewal and multidirectional differentiation into osteocyte, chondrocyte and adipocytes.

Although, definitive data supporting the stemness of this heterogeneous cell population were not provided, it has been hypothesized that a stromal stem cell population may be present in the stromal component of the bone marrow microenvironment and in that of other tissues; thus, the term “mesenchymal stem cell” should be restricted to this population of mesenchymal cells

Please revise the correct markers that characterize the MSCs (lines 66-67) (http://dx.doi.org/ 10.1002/stem.1681 https://doi.org/10.1007/s12015-021-10231-w, https://doi.org/10.1016/j.jcyt.2020.07.003).

=> Responses: We truly appreciate this valuable comment. We had reviewed the papers that you commented on to revise our manuscript. We have deleted the previous statement as “Mesenchymal stem cell (MSC), also named as mesenchymal stromal cell, is one of the important multipotent, … lineages: osteocytes, adipocytes, myocytes, and chondrocytes”.

Instead, we added a new paragraph as suggested, describing the different definitions and the correct markers by the ISCT criteria in the introduction section on page 2, lines 72-86 in the marked version of the revised manuscript as follows.

The acronym “MSC” usually refers to two different words in different literature: mesenchymal stem cell or mesenchymal stromal cell (Bianco, 2014; Galderisi et al., 2021). The two words have different nomenclature and definition. The former refers to stem cells in the skeletal tissue which are capable of both self-renewal and differentiation. The latter refers to the various cells in the stromal tissues, including multipotent mesenchymal stromal cells (with the same acronym “MSC”), progenitors, and differentiated mature cells. The International Society for Cellular Therapy (ISCT) recommended a minimal criteria to characterize multipotent mesenchymal stromal cells: 1. Plastic-adherent in standard culture conditions, 2. Positive for CD105, CD73 and CD90, and negative of CD45, CD34, CD14 or CD11b, CD79a or CD19 and HLA-DR surface molecules, 3. The differentiation ability into mesodermal lineages: osteoblasts, adipocytes, myocytes, and chondrocytes in vitro (Dominici et al., 2006). The multipotent mesenchymal stromal cells have the capability of self-renewal and multidirectional differentiations as mesenchymal stem cells do (Horwitz et al., 2005; Lv et al., 2014; Pittenger et al., 2019). Therefore, the literature included in this article was from studies of either mesenchymal stem cells or multipotent mesenchymal stromal cells.

2. Line 185 please change surface proteins with Surface proteins

=> Responses: We truly appreciate this valuable comment. We had corrected this mistake on page 7, line 230 in the marked version of the revised manuscript.

3. As suggestion, a curated list of experimentally-supported associations between lncRNAs and diseases can be found at https://www.cuilab.cn/lncrnadisease

=> Responses: We truly appreciate this valuable comment. According to the database from website https://www.cuilab.cn/lncrnadisease, there is a novel intronic antisense lncRNA named TrAnscript Predicting Survival in AKI (Tap-SAKI) which had been investigated in the AKI field. We have rewrited a new paragraph on page 13, lines 420-429 in the marked version of the revised manuscript as follows.

Focusing on the lncRNA in the AKI field, there is increasing evidence of lncRNA involvement in AKI. Lorenzen et al. first identified a novel intronic antisense lncRNA, named TrAnscript Predicting Survival in AKI (TapSAKI), could be found in kidney biopsy samples, especially enriched in hypoxic TECs, and also circulated in the blood of AKI patients (Lorenzen et al., 2015). The baseline concentrations of circulating TapSAKI lncRNA increased with the disease severity and correlated with the 28-day survival rate. The TapSAKI lncRNA was up-regulated in lipopolysaccharide (LPS)-induced HK-2 cell injury model, and knockdown of TapSAKI lncRNA could diminish kidney cell injury through several signaling pathways, including miR-205/Interferon regulatory factor 3 (IRF3) pathways, miR-22/PTEN/TLR4/NF-κB pathway (Han et al., 2021; Shen et al., 2019).

References:

Bianco, P. (2014). “Mesenchymal” Stem Cells. Annual Review of Cell and Developmental Biology 30, 677-704.

Dominici, M., Le Blanc, K., Mueller, I., Slaper-Cortenbach, I., Marini, F., Krause, D., Deans, R., Keating, A., Prockop, D., and Horwitz, E. (2006). Minimal criteria for defining multipotent mesenchymal stromal cells. The International Society for Cellular Therapy position statement. Cytotherapy 8, 315-317.

Galderisi, U., Peluso, G., and Di Bernardo, G. (2021). Clinical Trials Based on Mesenchymal Stromal Cells are Exponentially Increasing: Where are We in Recent Years? Stem Cell Reviews and Reports.

Han, X., Yuan, Z., Jing, Y., Zhou, W., Sun, Y., and Xing, J. (2021). Knockdown of lncRNA TapSAKI alleviates LPS-induced injury in HK-2 cells through the miR-205/IRF3 pathway. Open Med (Wars) 16, 581-590.

Horwitz, E.M., Le Blanc, K., Dominici, M., Mueller, I., Slaper-Cortenbach, I., Marini, F.C., Deans, R.J., Krause, D.S., and Keating, A. (2005). Clarification of the nomenclature for MSC: The International Society for Cellular Therapy position statement. Cytotherapy 7, 393-395.

Lorenzen, J.M., Schauerte, C., Kielstein, J.T., Hübner, A., Martino, F., Fiedler, J., Gupta, S.K., Faulhaber-Walter, R., Kumarswamy, R., Hafer, C., et al. (2015). Circulating long noncoding RNATapSaki is a predictor of mortality in critically ill patients with acute kidney injury. Clin Chem 61, 191-201.

Lv, F.-J., Tuan, R.S., Cheung, K.M.C., and Leung, V.Y.L. (2014). Concise Review: The Surface Markers and Identity of Human Mesenchymal Stem Cells. STEM CELLS 32, 1408-1419.

Pittenger, M.F., Discher, D.E., Péault, B.M., Phinney, D.G., Hare, J.M., and Caplan, A.I. (2019). Mesenchymal stem cell perspective: cell biology to clinical progress. npj Regenerative Medicine 4, 22.

Shen, J., Liu, L., Zhang, F., Gu, J., and Pan, G. (2019). LncRNA TapSAKI promotes inflammation injury in HK-2 cells and urine derived sepsis-induced kidney injury. J Pharm Pharmacol 71, 839-848.

Reviewer 3 Report

The review paper by Lee et al summarizes very well the current status on the molecular mechanisms of MSC therapy of AKI. I support such work and have only minor criticisms and suggestions.

Criticisms:

Page 2, line 66: ….., expression of MSC markers such as SSEA-4, EPO-R,……and then the reference to the positioning statement by Dominici et al. Please revise. The positioning statement refers to the expression of CD73,90,105 and the failure of CD45,11b,14,19HLA-DR-expression. Expression of SSEA-4 or EPO-R is not mentioned throughout this manuscript.

The authors provide an overview of MSC-Transplantation and EV-infusion studies, but only marginally discuss the topic of conditioned medium (with Evs in it, but not separated Evs!), and also pretreatment regimens to improve therapeutic potential. This could be reworked.

Table 1: Since Tab. 1 also includes studies with EV types that are not precisely characterized (NM), a study with an infusion of conditioned medium (CM) could also be a good fit here. CM and preconditioned CM contain large amounts of EVs. The study by Overath et al using a conditioned medium of adipose-derived MSCs in a cisplatin-induced AKI model is missing and could be added (Overath JM et al. Short-term preconditioning enhances the therapeutic potential of adipose-derived stromal/stem cell-conditioned medium in cisplatin-induced acute kidney injury. Exp Cell Res. 2016;342(2):175-83).

Furthermore, the tracking of transplanted MSCs in an AKI model could be discussed in the general part of the manuscript (Schubert R, et al. Tracking of Adipose-Derived Mesenchymal Stromal/Stem Cells in a Model of Cisplatin-Induced Acute Kidney Injury: Comparison of Bioluminescence Imaging versus qRT-PCR. Int J Mol Sci. 2018;19(9)).

Finally, the authors could write a few sentences in the conclusion part about how infused EVs may overcome the glomerular barrier or what paths there are to the (proximal) tubule to stimulate regeneration in this part of the nephron.

Author Response

Point-by-point responses to the reviewer (Manuscript ID: ijms-1344005)

Reviewer #3:

The review paper by Lee et al summarizes very well the current status on the molecular mechanisms of MSC therapy of AKI. I support such work and have only minor criticisms and suggestions.

Criticisms:

1. page 2, line 66: ….., expression of MSC markers such as SSEA-4, EPO-R,……and then the reference to the positioning statement by Dominici et al. Please revise. The positioning statement refers to the expression of CD73,90,105 and the failure of CD45,11b,14,19HLA-DR-expression. Expression of SSEA-4 or EPO-R is not mentioned throughout this manuscript.

=> Responses: We truly appreciate this valuable comment. We have clarified our confusion and deleted the previous statement. Instead, we added a new paragraph as suggested, describing the different definitions and the correct markers by the ISCT criteria in the introduction section on page 2, lines 72-86 in the marked version of the revised manuscript as follows.

The acronym “MSC” usually refers to two different words in different literature: mesenchymal stem cell or mesenchymal stromal cell (Bianco, 2014; Galderisi et al., 2021). The two words have different nomenclature and definition. The former refers to stem cells in the skeletal tissue which are capable of both self-renewal and differentiation. The latter refers to the various cells in the stromal tissues, including multipotent mesenchymal stromal cells (with the same acronym “MSC”), progenitors, and differentiated mature cells. The International Society for Cellular Therapy (ISCT) recommended a minimal criteria to characterize multipotent mesenchymal stromal cells: 1. Plastic-adherent in standard culture conditions, 2. Positive for CD105, CD73 and CD90, and negative of CD45, CD34, CD14 or CD11b, CD79a or CD19 and HLA-DR surface molecules, 3. The differentiation ability into mesodermal lineages: osteoblasts, adipocytes, myocytes, and chondrocytes in vitro (Dominici et al., 2006). The multipotent mesenchymal stromal cells have the capability of self-renewal and multidirectional differentiations as mesenchymal stem cells do (Horwitz et al., 2005; Lv et al., 2014; Pittenger et al., 2019). Therefore, the literature included in this article was from studies of either mesenchymal stem cells or multipotent mesenchymal stromal cells.

2. The authors provide an overview of MSC-Transplantation and EV-infusion studies, but only marginally discuss the topic of conditioned medium (with EVs in it, but not separated EVs!), and also pretreatment regimens to improve therapeutic potential. This could be reworked.

Table 1: Since Tab. 1 also includes studies with EV types that are not precisely characterized (NM), a study with an infusion of conditioned medium (CM) could also be a good fit here. CM and preconditioned CM contain large amounts of EVs. The study by Overath et al. using a conditioned medium of adipose-derived MSCs in a cisplatin-induced AKI model is missing and could be added (Overath JM et al. Short-term preconditioning enhances the therapeutic potential of adipose-derived stromal/stem cell-conditioned medium in cisplatin-induced acute kidney injury. Exp Cell Res. 2016;342(2):175-83).

=> Responses: We truly appreciate this valuable comment. We have reviewed the pre-conditioned medium therapy for AKI, rewrited a new paragraph, and revised Table 1. Please refer to the newly added paragraph “Furthermore, human MSC-conditioned media (MSC-CM) also possess regenerative properties for tissue injury due to MSC-secreted products, such as proteins, lipids, cytokines, or EVs, etc. Overath et al. found that MSC-CM from ADSCs preincubated in a hypoxic environment contains more protective factors and had better therapeutic effects for cisplatin-induced AKI mice than ordinary MSC-CM (Overath et al., 2016)” on page 4, lines 176-181, and Table 1 on page 5 in the marked version of the revised manuscript.

3. Furthermore, the tracking of transplanted MSCs in an AKI model could be discussed in the general part of the manuscript (Schubert R, et al. Tracking of Adipose-Derived Mesenchymal Stromal/Stem Cells in a Model of Cisplatin-Induced Acute Kidney Injury: Comparison of Bioluminescence Imaging versus qRT-PCR. Int J Mol Sci. 2018;19(9)).

=> Responses: We truly appreciate this valuable comment. We have added a new paragraph on page 3, lines 142-148 in the marked version of the revised manuscript as follows.

Schubert et al. had demonstrated the MSC migration and homing by tracking the adipose-derived MSC (ADSC) with bioluminescence imaging (BLI) versus quantitative reverse transcription polymerase chain reaction (qRT-PCR) in the cisplatin-induced AKI model (Schubert et al., 2018). They detected the labeled Luc-specific mRNA in the injured kidney tissue by using qRT-PCR; however, they only detected Luc+-ADSCs in the lung, but not in the kidney, suggesting that Luc-based qRT-PCR might be a better tool than BLI to track the transplanted MSCs.

References:

Bianco, P. (2014). “Mesenchymal” Stem Cells. Annual Review of Cell and Developmental Biology 30, 677-704.

Dominici, M., Le Blanc, K., Mueller, I., Slaper-Cortenbach, I., Marini, F., Krause, D., Deans, R., Keating, A., Prockop, D., and Horwitz, E. (2006). Minimal criteria for defining multipotent mesenchymal stromal cells. The International Society for Cellular Therapy position statement. Cytotherapy 8, 315-317.

Galderisi, U., Peluso, G., and Di Bernardo, G. (2021). Clinical Trials Based on Mesenchymal Stromal Cells are Exponentially Increasing: Where are We in Recent Years? Stem Cell Reviews and Reports.

Horwitz, E.M., Le Blanc, K., Dominici, M., Mueller, I., Slaper-Cortenbach, I., Marini, F.C., Deans, R.J., Krause, D.S., and Keating, A. (2005). Clarification of the nomenclature for MSC: The International Society for Cellular Therapy position statement. Cytotherapy 7, 393-395.

Lv, F.-J., Tuan, R.S., Cheung, K.M.C., and Leung, V.Y.L. (2014). Concise Review: The Surface Markers and Identity of Human Mesenchymal Stem Cells. STEM CELLS 32, 1408-1419.

Overath, J.M., Gauer, S., Obermüller, N., Schubert, R., Schäfer, R., Geiger, H., and Baer, P.C. (2016). Short-term preconditioning enhances the therapeutic potential of adipose-derived stromal/stem cell-conditioned medium in cisplatin-induced acute kidney injury. Exp Cell Res 342, 175-183.

Pittenger, M.F., Discher, D.E., Péault, B.M., Phinney, D.G., Hare, J.M., and Caplan, A.I. (2019). Mesenchymal stem cell perspective: cell biology to clinical progress. npj Regenerative Medicine 4, 22.

Schubert, R., Sann, J., Frueh, J.T., Ullrich, E., Geiger, H., and Baer, P.C. (2018). Tracking of Adipose-Derived Mesenchymal Stromal/Stem Cells in a Model of Cisplatin-Induced Acute Kidney Injury: Comparison of Bioluminescence Imaging versus qRT-PCR. Int J Mol Sci 19.